# Experimental investigation of the influence of friction, surface roughness and material hardness on the external load factor in threaded joints

**Van Thuy Tran**[ID][1*], **Huu Loc Nguyen**[2,3]

**1** Faculty of Engineering-Technology, Pham Van Dong University, Quang Ngai Province, Vietnam,
**2** Department of Machine Design, Faculty of Mechanical Engineering, Ho Chi Minh City University of Technology (HCMUT), Ho Chi Minh City, Vietnam, **3** Vietnam National University Ho Chi Minh City, Ho Chi Minh City, Vietnam

\* tvthuy@pdu.edu.vn

## Abstract

Threaded joints, particularly bolted joints, are critical components in the design and fabrication of mechanical systems due to their high strength and ease of disassembly. Their widespread application spans across structural engineering, transportation, bridge construction, and industrial machinery. The external load factor ($\chi$), which characterizes the load-sharing behavior between the bolt and the clamped components, plays a vital role in assessing joint performance and reliability. This study presents an experimental investigation into the effects of friction coefficient, surface roughness, and material hardness of the clamped plates on the external load factor in threaded joints. A second-order regression model is developed using the Box–Wilson design method to quantitatively describe the influence of these parameters. The results reveal that all three factors significantly affect $\chi$, with material hardness showing the strongest influence, $\chi$ decreases as hardness increases. Surface roughness exhibits a nonlinear effect, while the friction coefficient also has a notable impact. An optimization was carried out to identify the parameter combination that meets the target external load factor of $\chi = 0.2501$. The optimal conditions include a friction coefficient $\mu = 0.1809$, surface roughness $Ra = 3.5181\ \mu m$, and material hardness of $HB = 216.859$. The resulting composite desirability of 0.9996 confirms the model's predictive accuracy and the high reliability of the joint under the specified conditions. These findings allow better design of reliable bolted joints in engineering applications.

## Introduction

Threaded joints, particularly bolted joints, are critical components in the design and fabrication of mechanical systems due to their versatility and ease of assembly and

**Data availability statement:** All data underlying the findings of this study are fully available within the paper and its Supporting Information files. The experimental dataset used for the regression equation (22) is provided as Supporting Information (S1 File) in XLSX format. No additional data are required to replicate the results reported in this study.

**Funding:** The author(s) received no specific funding for this work.

**Competing interests:** The authors have declared that no competing interests exist.

disassembly. They play a critical role in maintaining structural integrity in various applications, including construction, transportation, and mechanical manufacturing [1–3]. The performance and reliability of bolted joints are directly influenced by how external loads are distributed between the bolt and the clamped components. Specifically, in a threaded joint under an initial preload, the bolt is subjected to tensile force while the clamped components are compressed. When an external load is applied, within the limits .where neither separation nor slippage occurs at the interface, the bolt elongates by an additional amount Δl, while the compressive deformation of the clamped parts decreases by the same amount [4,5]. In other words, only a portion χF of the external force F contributes to further elongation of the bolt, while the remaining portion (1 − χ)F reduces the compressive deformation of the clamped parts. The coefficient χ is referred to as the external load factor. This load distribution depends on the relative stiffness of the bolt and the clamped members. In other words, it is governed by the external load factor itself [6–8]. Accurate determination of this factor is essential for evaluating the load capacity of the joint, especially in applications requiring high safety margins or those subjected to cyclic loading.

In classical models, the external load factor is typically determined through stiffness analysis by representing the bolt and clamped components with equivalent spring systems [9–11]. However, these models assume ideal assembly conditions, such as perfect contact surfaces, absence of friction, and isotropic material properties, which often result in significant discrepancies when applied to real-world scenarios. Recent studies have extended the theoretical models by incorporating additional factors such as thread friction, surface friction and variability in tightening torque to improve the accuracy of load distribution predictions [12–16]. For example, these studies have analyzed the influence of combined friction coefficients on load distribution in parallel bolt assemblies. However, accurately determining these parameters in practice is often challenging due to significant variations caused by environmental conditions, material properties, and assembly procedures. An alternative approach involves using experimental or semi-empirical models to establish relationships between the external load factor and input parameters. Design of Experiments (DoE) methods, such as Box–Wilson or Taguchi, have been employed to simultaneously investigate the effects of multiple variables including surface roughness, tightening torque, material hardness, and lubrication [17–21]. Previous studies have demonstrated that surface characteristics and tightening methods significantly affect the initial preload and external load factor in joints. Furthermore, the development of experimental devices for measuring the external load factor in laboratory settings has attracted considerable attention. Force sensors integrated within bolts or placed on clamped surfaces enable direct measurement of bolt response under external loads [22–24]. Data obtained from such measurement systems can be used to calibrate theoretical models and support the construction of more accurate regression models. However, to date, there remains a lack of systematic studies that comprehensively evaluate the influence of assembly conditions, such as friction, surface roughness, and material hardness, on the external load factor through both repeated experimental measurements and quantitative modeling. Notably, no research has yet combined

second-order regression models with experimental measurement devices to identify key influencing factors and determine optimal configurations aimed at minimizing the external load factor.

Numerous classical theoretical models have been developed to estimate the external load factor; however, most of these models rely on idealized assumptions and do not account for the influence of actual assembly conditions. In practice, factors such as thread friction, contact surface friction, surface roughness, and material hardness of the clamped components significantly affect the load transfer mechanism within the joint. Yet, the extent of their individual and interactive effects has not been comprehensively quantified. The advancement of modern measurement devices now enables direct measurement of the external load factor through sensors and specialized instrumentation. The implementation of direct force sensor measurements further strengthens the study's practical significance by allowing precise, in-situ evaluation of joint behavior under realistic assembly and loading conditions.

This study aims to address the existing gap by employing a specialized external load factor measurement device to conduct experimental investigations based on the Box–Wilson design method, focusing on three primary factors: friction conditions, surface roughness and material hardness. A second-order regression model is developed to serve as the foundation for evaluating and optimizing threaded joints under realistic assembly conditions. The experimental results are used to construct a regression equation that captures the nonlinear effects and interactions among these parameters on the external load factor. The model not only reflects the individual roles of each factor but also enables the identification of the optimal assembly configuration to minimize the external load factor, thereby enhancing the reliability and fatigue life of the joints.

## Theoretical background

### Load decomposition and equivalent moment

In practice, the load acting on a threaded joint may have an arbitrary direction. The joint is assumed to be subjected to a load of arbitrary direction lying in the YY symmetry plane, which is defined as the vertical symmetry plane passing through the bolt axis and perpendicular to the clamped surface.

The external load F can be resolved into two components: $F_V$, which acts perpendicular to the clamped surface and $F_H$, which acts parallel to it, as illustrated in Fig 1. When these force components are transferred to the centroid C of the bolt group, they generate an equivalent moment M, determined according to Equation (1), which contributes to the overall loading condition of the joint. In most practical configurations, the bolt group exhibits two mutually perpendicular symmetry axes, with the centroid C located at their intersection.

$$M = F_H l_2 - F_V l_1 \tag{1}$$

When subjected to the moment M, the joint tends to rotate about an axis lying within the clamped plane. According to the principle of minimum moment of inertia, the rotation axis can be assumed to coincide with the symmetry axis XX passing through the centroid C of the bolt group, since the resisting moment for this axis is the smallest. However, this assumption is valid only when the bolt preload is sufficiently high to prevent separation of the clamped surfaces.

The load component $F_V$ and the moment M tend to separate the clamped surfaces, whereas the component $F_H$ causes sliding between the clamped plates. The load $F_V$ and moment M are divided into two parts: $F_b$ and $M_b$, acting on the bolts; and $F_m$ and $M_m$, acting on the clamped members, as expressed in Equation (2), where χ denotes the external load factor.

$$\begin{cases} F_b = \chi F_V \\ F_m = (1-\chi) F_V \\ M_b = \chi M \\ M_m = (1-\chi) M \end{cases} \tag{2}$$

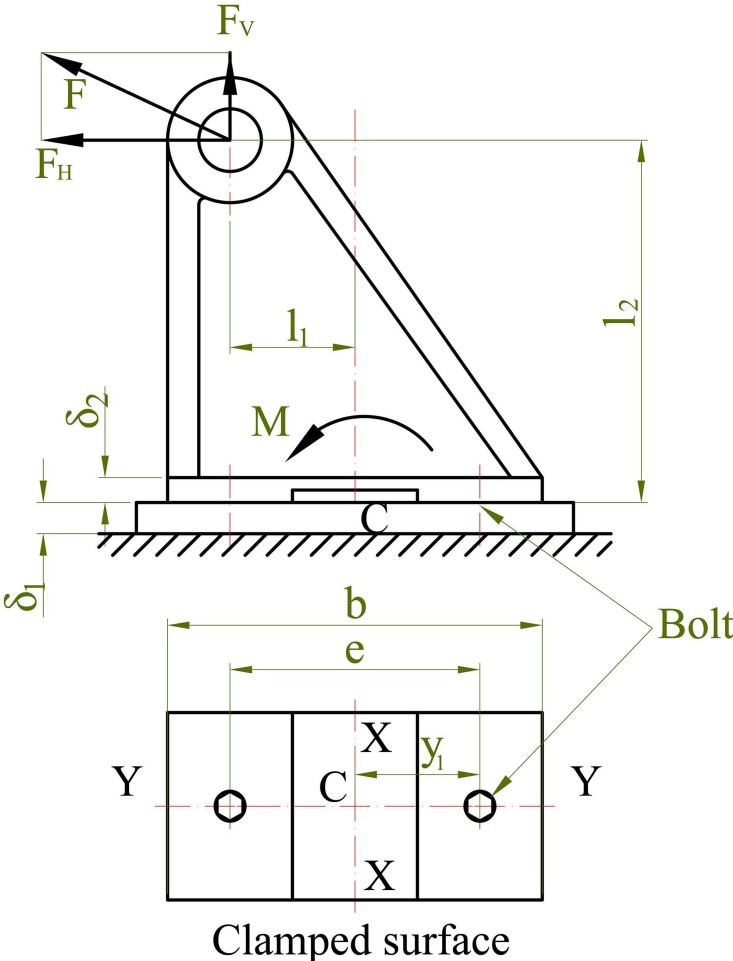

**Fig 1. Bolted joint load model.**

## Determination of the tightening force

To prevent separation of the clamped members and to avoid slippage at the interface, the bolts must be tightened with an appropriate preload force V. When the preload and external loads are simultaneously applied, the combined effect of the horizontal load $F_H$ and the moment M produce a nonuniform distribution of tensile stresses among the bolts. The resulting stress state in the clamped section can be divided into several distinct components, as illustrated in Fig 2.

## Prevention of separation

Before the external load $F_v$ is applied, the joint is subjected to bearing stress ($\sigma_v$) generated by the initial tightening force V applied to the bolts. The stress distribution resulting from this initial tightening is illustrated in Fig 2(b), and its magnitude is determined according to Equation (3).

$$\sigma_V = \frac{zV}{A_m}$$

(3)

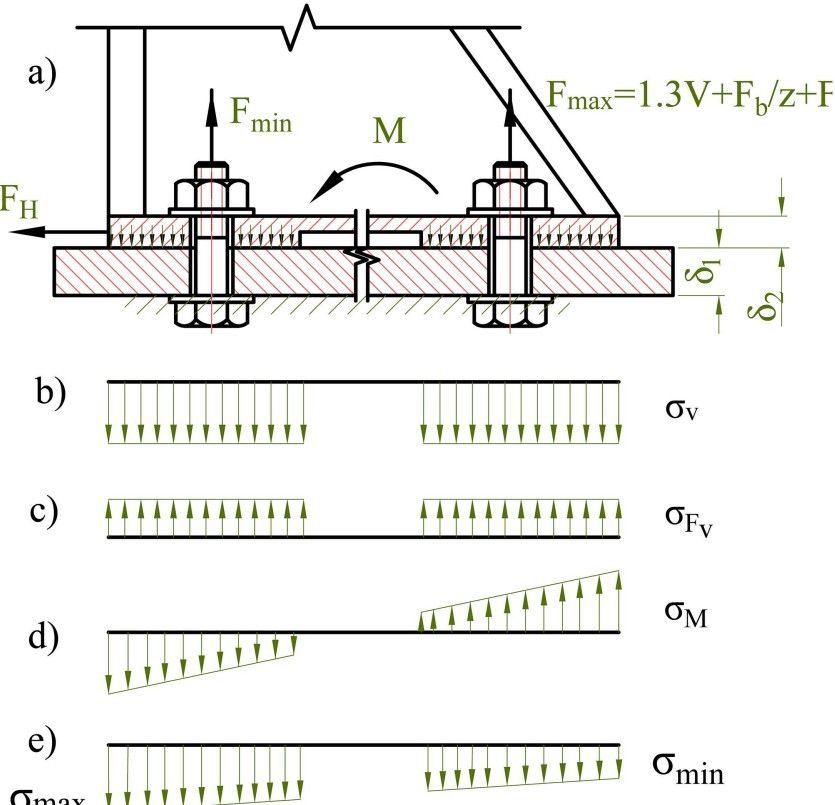

**Fig 2. Force and stress distributions in a bolted joint.** (a) Free-body diagram showing external load $F_H$, moment M and bolt forces; (b–e) distributions of normal stress components $\sigma_V$, $\sigma_F$, $\sigma_M$, and resultant stresses $\sigma_{max}$, $\sigma_{min}$.

Where $A_m$ is the effective contact area between the clamped surfaces, V is the initial tightening force of each bolt, and z is the total number of bolts in the joint.

When the external load acts on the joint, the bearing stress on the contact surface decreases by an amount $\sigma_F$, corresponding to the load $F_m$ transmitted through the clamped members. The stress distribution resulting from this load is illustrated in Fig 2(c), and its magnitude is determined according to Equation (4).

$$\sigma_F = \frac{F_m}{A_m}$$

(4)

For relatively rigid clamped plates, the bending deformation within the joint can be approximated as linear. Accordingly, within the elastic range, the corresponding stress distribution on the interface also follows a linear pattern. The diagram in Fig 2(d) illustrates the stress distribution caused by the bending moment $M_m$. The maximum value of this stress is determined according to Equation (5).

$$\sigma_M = \frac{M_m}{W_m} = \frac{M_m y_c}{J_m}$$

(5)

Where $W_m$ is the bending section modulus of the clamped surface, $J_m$ is the moment of inertia of the clamped surface with respect to the tilting axis XX, and $y_c$ is the distance from the outermost point to the tilting axis XX.

The maximum and minimum total stresses on the contact surface, resulting from the combined effect of the tightening force and the external load, are illustrated in Fig 2(e) and determined according to Equation (6).

$$\sigma_{\substack{\max \\ \min}} = \sigma_V - \sigma_F \pm \sigma_M = \frac{zV}{A_m} - \frac{F_m}{A_m} \pm \frac{M_m}{W_m} \tag{6}$$

In most practical cases, the contact surface area is much larger than the total area of the bolt holes. Therefore, it can be assumed that $A_m \approx A$ and $W_m \approx W$, where A and W are the area and the bending section modulus of the solid (unperforated) section, respectively. Under this assumption, Equation (6) can be simplified to:

$$\sigma_{\substack{\max \\ \min}} = \sigma_V - \sigma_F \pm \sigma_M = \frac{zV}{A} - \frac{F_m}{A} \pm \frac{M_m}{W} \tag{7}$$

To prevent separation of the joint, the minimum contact stress must satisfy the condition $\sigma_{\min} > 0$, which is equivalent to:

$$\sigma_{\min} = \frac{zV}{A} - \frac{F_m}{A} - \frac{M_m}{W} = \frac{zV}{A} - \frac{(1-\chi)F_V}{A} - \frac{(1-\chi)M}{W} > 0 \tag{8}$$

From Equation (8), the required tightening force V for each bolt is determined according to Equation (9).

$$V > \frac{1}{z}\left(F_V + \frac{MA}{W}\right)(1-\chi) \tag{9}$$

For safety purposes, by multiplying with the safety factor k, Equation (9) can be rewritten as follows:

$$V = \frac{k}{z}\left(F_V + \frac{MAy_c}{J}\right)(1-\chi) \tag{10}$$

where k is the safety factor used to prevent separation of the joint, typically ranging from 1.3 to 2.0.

**Prevention of slippage**

For a joint using bolts with clearance between the hole and the bolt shank, in order to prevent the clamped plates from slipping, the force $F_H$ is resisted by the frictional force generated on the contact surface. The joint will not slip if $F_H$ is less than the maximum frictional force, which is:

$$f(zV - F_m) > F_H \tag{11}$$

For safety purposes, by multiplying with the safety factor k, Equation (11) can be rewritten as follows:

$$f(zV - F_m) = kF_H \tag{12}$$

where k is the safety factor used to prevent separation of the joint, typically ranging from 1.3 to 2.0.

From Equation (12), the required tightening force V for each bolt is determined according to Equation (13).

$$V = \frac{kF_H + fF_m}{fz} = \frac{kF_H + (1 - \chi) fF_V}{fz} \tag{13}$$

## Equivalent tensile force in bolts

To ensure proper joint performance, the tightening force V is selected as the larger of the two values obtained from Equations (10) and (13). In addition to the tightening force V, each bolt is subjected to forces resulting from $F_b$ and the moment $M_b$ under external loading. When the force $F_b$ acts, each bolt in the group carries a force of $F_b/z$. Due to the action of the moment $M_b$, the bolts experience uneven forces; the bolt in the outermost left row, at the maximum distance $y_1$ from the tilting axis XX, experiences the largest tensile force.

Let $F_{M1}$, $F_{M2}$… denote the forces generated by the moment $M_b$ on the bolts at distances $y_1$, $y_2$,… from the axis XX. These forces are determined according to Equation (14).

$$F_{M2} = F_{M1} \frac{y_2}{y_1}; \; F_{M3} = F_{M1} \frac{y_3}{y_1}; \tag{14}$$

The equilibrium condition for the bolt group is expressed by Equation (15):

$$M_b = z_1 F_{M1} y_1 + z_2 F_{M2} y_2 + \cdots = F_{M1} \sum z_i \frac{y_i^2}{y_1} \tag{15}$$

where $z_i$ represents the number of bolts positioned at the same distance $y_i$ from the tilting axis.

From this relationship, the tensile force $F_{M1}$ generated by the moment $M_b$ on the bolt farthest from the tilting axis XX is determined according to Equation (16).

$$F_{M1} = \frac{M_b y_1}{\sum z_i y_i^2} = \frac{\chi M y_1}{\sum z_i y_i^2} \tag{16}$$

The total force acting on the bolt subjected to the maximum load is then given by Equation (17).

$$F_{\max} = V + \frac{F_b}{z} + F_{M1} \tag{17}$$

If the torsional stress component caused by the thread moment is considered, the tightening force V should be multiplied by 1.3. Consequently, the equivalent design tensile force $F_{td}$ is determined according to Equation (18).

$$F_{td} = 1,3V + \frac{F_b}{z} + F_{M1} = V_0 + \frac{\chi F_V}{z} + \frac{\chi M y_1}{\sum z_i y_i^2} \tag{18}$$

Where $V_0$ is the initial tightening force, with a value of 1.3V.

## Determination of the external load factor

In this experiment, the equivalent tensile force $F_{td}$ is derived from direct measurement. Accordingly, it is denoted as $V_{tn} = F_{td}$:

$$V_{tn} = F_{td} = V_0 + \chi \left( \frac{F_V}{z} + \frac{My_1}{\sum z_i y_i^2} \right)$$

$$(19)$$

The external load is then applied incrementally using a hydraulic cylinder with magnitudes $F_1$, $F_2$, … $F_N$, producing N corresponding measured values $V_{tn1}$, $V_{tn2}$,…, $V_{tn}$. Consequently, the relationship can be expressed as:

$$V_{tn1} = V_0 + \chi \left( \frac{F_{V1}}{z} + \frac{M_1 y_1}{\sum z_i y_i^2} \right) = V_0 + \chi \left( \frac{F_{V1}}{z} + \frac{F_{H1} l_1 \pm F_{V1} l_2}{2e} \right)$$

$$V_{tn2} = V_0 + \chi \left( \frac{F_{V2}}{z} + \frac{M_2 y_1}{\sum z_i y_i^2} \right) = V_0 + \chi \left( \frac{F_{V2}}{z} + \frac{F_{H2} l_1 \pm F_{V2} l_2}{2e} \right)$$

$$V_{tnN} = V_0 + \chi \left( \frac{F_{VN}}{z} + \frac{M_N y_1}{\sum z_i y_i^2} \right) = V_0 + \chi \left( \frac{F_{VN}}{z} + \frac{F_{HN} l_1 \pm F_{VN} l_2}{2e} \right)$$

The external load factor $\chi$ is defined according to Equation (20).

$$\chi_i = \frac{V_{tni} - V_{tn1}}{\left( \frac{F_{Vi} - F_{V1}}{z} + \frac{(F_{Hi} - F_{H1}) l_1 + (F_{Vi} - F_{V1}) l_1}{2e} \right)}$$

$$(20)$$

At that time, the average external load factor over N measurements is given by Equation (21).

$$\chi = \frac{\chi_1 + \chi_2 + ... + \chi_{N-1}}{(N-1)}$$

$$(21)$$

**Principle of measuring the external load factor**

The measuring system is designed to determine the external load factor ($\chi$) by simultaneously recording the bolt tightening force and the applied external load. The schematic diagram of the measurement principle is shown in Fig 3 [8].

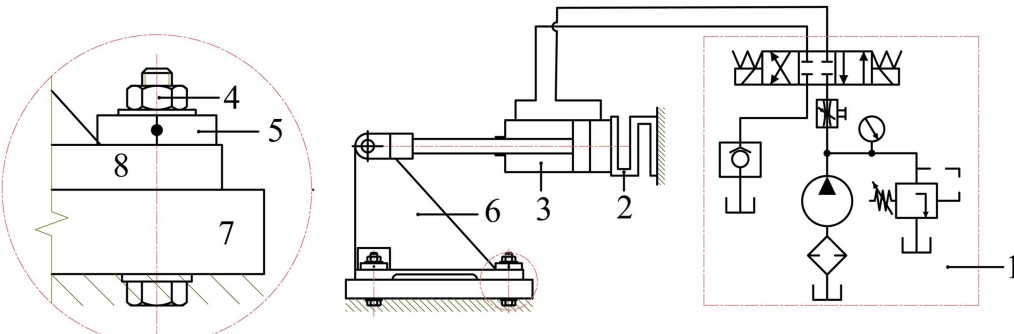

**Fig 3. Schematic diagram of the system.**

*1. Hydraulic system; 2. Loadcell measuring the cylinder force; 3. Hydraulic cylinder generating force; 4. Bolt joint; 5. Loadcell measuring bolt tightening force; 6. Frame structure; 7,8. Clamping plates*

To determine the external load factor, the bolt joint (4) is first tightened with an initial preload force $V_0$ to prevent slipping and separation of the joint. The initial tightening force is measured by the loadcell sensor (5). Subsequently, the hydraulic system (1) actuates a hydraulic cylinder (3) to apply an arbitrary external load to the bolt joint (4). The force generated by the cylinder is measured by the loadcell sensor (5). Under the action of increasing external load, the tightening force of the bolt joint gradually rises to a new value V. As the external load increases, the bolt tightening force rises correspondingly to V. Based on the simultaneously recorded values $F_i$ and $V_{tni}$, the external load factor $\chi$ is calculated using Equations (20) and (21).

## Experimental methodology

### Experimental setup

The experimental setup for determining the external load factor is shown in Fig 4.

*1. External force display meter; 2. Electrical control cabinet; 3. Bolt tightening force display meter; 4. Hydraulic system; 5,11. Clamping plates; 6. Support frame; 7. Force applicator arm; 8. Tilt angle adjuster (α); 9. Bolt; 10. Loadcell measuring bolt tightening force*

The test specimen consists of two steel plates joined by an M16 bolt through a circular hole, subjected to axial loading from the hydraulic cylinder. Two types of loadcells are used: (i) S-type loadcell (Zemic B3G, 0–20 kN, stainless steel) measures the external load; (ii) Donut through-hole loadcell (Forsentek FBDA, 0–20 kN) measures the bolt tightening force.

The sensors have received factory calibration with full-scale accuracy ±0.05%, repeatability ±0.02%, and linearity error below ±0.1%, ensuring reliable and stable measurements throughout the loading process.

The configuration and installation method of the donut-type loadcell are illustrated in Fig 5. In the technical diagrams (Figs 4 and 5), geometric dimensions are presented in millimeters (mm), while force measurements are expressed in kilonewtons (kN). The sensor outputs are transmitted to a data acquisition system and displayed through a force indicator, enabling continuous recording of variations in bolt tightening force under the applied axial load.

*1. Bolt M16; 2. Loadcell measuring bolt tightening force; 3,4. Clamping plates.*

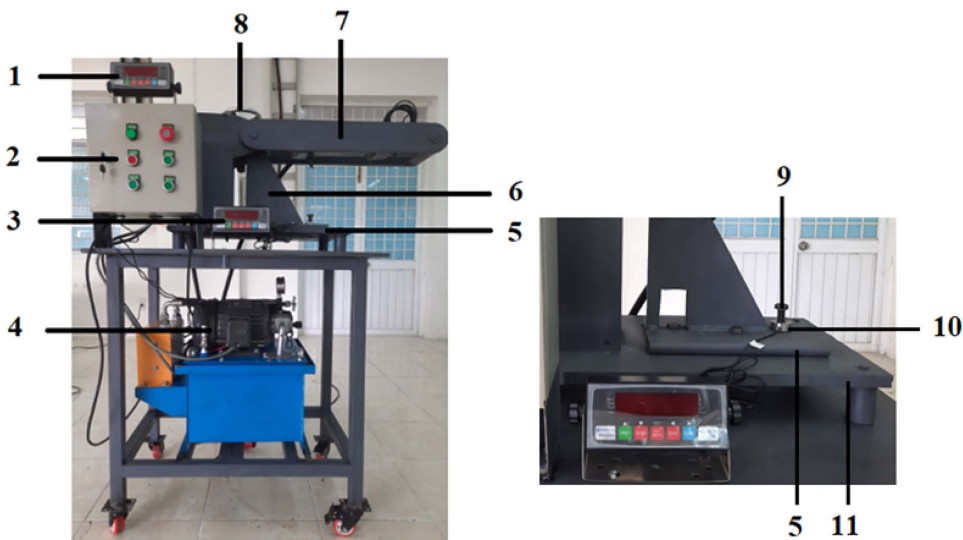

**Fig 4. Experimental Setup.**

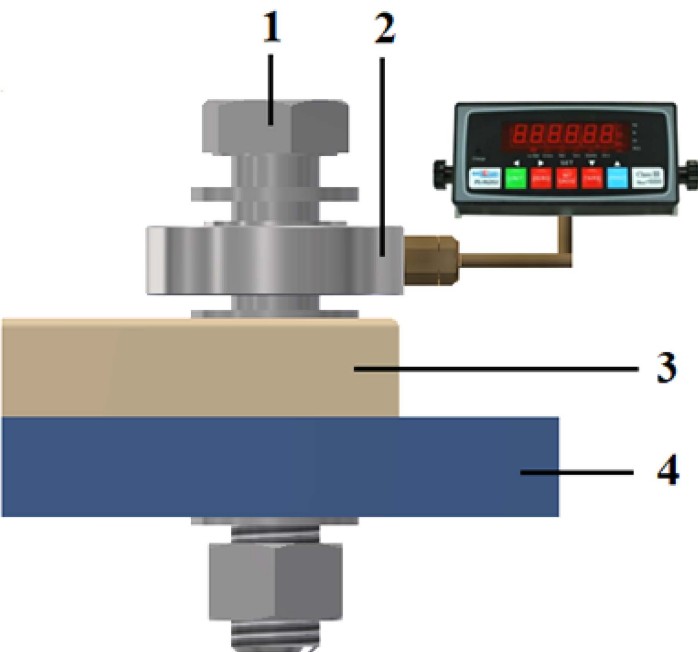

**Fig 5. Installation of donut loadcell for bolt preload measurement.**

## Design of experiment (DOE)

The three input factors, including friction, surface roughness and material hardness are selected to investigate their influence on the external load factor ($\chi$) in threaded joints. The selection of these factors is based not only on their significant impact on load distribution within the joint but also on the fact that they represent key characteristics of assembly conditions in practical mechanical manufacturing.

The combined friction coefficient ($\mu$) between contacting surfaces in the bolted joint, including friction at the threads and under the bolt head, plays a critical role in converting the tightening torque into clamping preload and strongly influences the external load factor ($\chi$). During bolt tightening, the tightening torque is not fully converted into axial preload due to significant losses caused by friction. Therefore, changes in the friction coefficient alter the achievable clamping force, resulting in different load distributions between the bolt and the joint components. According to technical references and ISO 16047 standards, the friction coefficient at contacting surfaces in bolted joints typically varies from 0.1 to 0.3 depending on surface conditions and lubrication status. Specifically, $\mu = 0.1$ represents well-lubricated conditions, commonly found when using standard lubricating oil or assembly grease, whereas $\mu = 0.3$ corresponds to dry or high-friction conditions, often encountered on unlubricated surfaces, rough surfaces or worn areas. Selecting these two distinct levels ensures representativeness of the typical friction range in practice and facilitates the construction of a quadratic regression model in experimental design.

Surface roughness (Ra) is an important factor affecting the actual contact between mating surfaces in a bolted joint, thereby influencing the friction coefficient and the clamping force within the joint. The rougher the surface, the smaller the actual contact area, resulting in higher friction and greater variation in clamping force during tightening, which alters the external load factor ($\chi$). Therefore, surface roughness is a practically significant parameter that needs to be investigated. In this study, two roughness levels were selected: Ra = 1.6 μm, representing finely machined surfaces such as ground or finely turned surfaces with low friction, commonly used in precision assemblies; and Ra = 6.3 μm, representing rough

surfaces that are either unfinished or worn, which increases contact friction. The surface roughness of the specimens was measured using a contact-type surface roughness tester, the SJ-210 – Series 178 (Mitutoyo, Japan), in accordance with ISO 1997 standards. The device employs a diamond-tipped stylus with a low contact force (~0.75 mN) to record the surface profile over a specified measuring length. Each specimen was measured at three different positions, and the average value was taken to ensure representative input data. The instrument's measuring range for Ra is 25 µm (±12.5 µm), with an accuracy of ±0.002 µm.

Although Ra may influence frictional behaviors, µ and Ra were controlled through different physical mechanisms in this study. Ra was adjusted solely through machining processes, whereas µ was adjusted through lubrication state. Friction in bolted joints depends on several factors beyond surface geometry, such as lubricant thickness, surface chemistry, and contamination, meaning that µ cannot be predicted from Ra alone. Consequently, µ and Ra are treated as independent variables in accordance with DOE principles, enabling independent quantification of geometric (Ra) and tribological (µ) effects, identification of interaction effects between Ra and µ and a more comprehensive understanding of how assembly conditions influence χ.

The material hardness of the Clamping plates (HB) is one of the key factors directly affecting the distribution of external forces in the system. When the joint is subjected to external loads, the load is not only applied to the bolt but also distributed through the contact areas of the joint components. In cases where the material has low hardness, greater contact deformation occurs, increasing the load transferred to the bolt and leading to a higher external load factor (χ). Conversely, if the material has higher hardness, the joint components share the load more effectively, reducing the load on the bolt and thus decreasing χ. Changing the joint material results in changes in relative hardness, thereby influencing the value of χ. Therefore, including material hardness as an input factors in the experimental design is necessary to quantitatively assess its impact on the external force distribution in the threaded joint. For experimental convenience, this study selected two representative hardness levels: aluminum alloy or mild steel (HB ≈ 120) and quenched steel (HB ≈ 250). This selection not only represents common hardness classes in mechanical manufacturing but also ensures clear control and comparison of experimental results. The Rockwell hardness tester (Model 574, Wilson, USA) was used to determine the material hardness of the clamped plates, in accordance with ASTM E18 and ISO 6508 standards. The device supports multiple Rockwell scales (A–V) with test loads ranging from 15 kg to 150 kg. Both dwell time and test cycles are adjustable. It accommodates samples up to 289 mm in height, with a throat depth of 175 mm. Measurement results are digitally displayed, and data can be exported via USB or RS-232 interfaces.

The experiments were based on a Rotatable Central Composite Design model (Box–Wilson), with the levels of each factor summarized in Table 1.

**Table 1. Design schema of input factors and their levels.**

| Variables or parameter | Factor symbol | | Variable range of input factors | | | | |
|---|---|---|---|---|---|---|---|
| | Actual factor | Code factor | level -α | level (−1) | level (0) | level (+1) | level +α |
| Friction coefficient,µ | µ | $x_1$ | 0.032 | 0.1 | 0.2 | 0.3 | 0.368 |
| Surface roughness Ra, µm | Ra | $x_2$ | 0.01 | 1.6 | 3.95 | 6.3 | 7.9 |
| Material hardness, HB | HB | $x_3$ | 75.67 | 120 | 185 | 250 | 294.33 |

A total of 20 experiments were conducted, each consisting of 7 repeated measurements to ensure statistical reliability. The experimental design includes 8 factorial points, 6 axial (star) points, and 6 center points, as defined by the Rotatable Central Composite Design. In the first experiment, the equivalent tensile force was measured for a joint configuration clamped by two M16 bolts. Based on the repeated trials, the standard deviation of the measured values of the load distribution coefficient (χ) was found to be within ±0.002. This small variation indicates a high level of measurement consistency and supports the statistical significance of the regression model. The geometric dimensions of the joint surface and the frame structure are e = 200 mm, $l_1$ = 100 mm and $l_2$ = 300 mm.

The experimental setup employs a hydraulic system capable of generating a maximum force of $F_{max} = 25\,$kN. According to the design criteria specified by Equations (10) and (13), the required preload force per bolt to ensure that the joint does not experience either slippage or separation is determined to be $V_0 \geq 15430\,$N. The bolts are pre-tightened according to this initial clamping force $V_0$. Subsequently, axial loads $F_1$, $F_2$, ... $F_N$, each less than the maximum load $F_{max}$, are incrementally applied via the hydraulic actuator. For each load step, the corresponding clamping force $V_{tn1}$, $V_{tn2}$,... $V_{tnN}$ is recorded using a through-hole load cell installed in the bolted joint. These experimental values are then used to compute the load transfer factor $\chi$ for the threaded connection under the specified initial preload condition. The measurement results and the calculated $\chi$ values for the two-bolt configuration are summarized in Table 2.

Following the same procedure, the results of the average external load factor of the 20 experimental runs are summarized in Table 3.

## Results and discussion

### The coefficients of regression equation

Based on the Box–Wilson experimental design model and data analysis using Minitab software, the coefficients of the regression equation were determined as follows:

$$\chi = 0.407 + 1.06\mu + 0.024Ra - 1.01 \times 10^{-3}HB - 9.6 \times 10^{-3}\mu Ra - 3.1 \times 10^{-4}\mu HB+$$

$$+ 2.29 \times 10^{-5}RaHB - 2.78\mu^2 - 4.14 \times 10^{-3}Ra^2 - 1.88 \times 10^{-6}HB^2 \tag{22}$$

After obtaining the regression equation (22), the next step is to evaluate the statistical significance of the regression coefficients and testing lack of fit in regression model.

### The testing of the individual regression coefficients

The significance of the regression coefficients is evaluated using the t-test for each coefficient, with a predetermined significance level (commonly 5%). Coefficients with P-values less than 0.05 are considered to have a significant effect on the output variable $\chi$, indicating statistical significance. Conversely, coefficients with P-values greater than 0.05 may be considered for removal if they do not substantially affect the model's accuracy. The significance of the regression coefficients is assessed using Minitab software.

After constructing the regression equation based on the Box-Wilson experimental design, the statistical analysis results are presented in Table 4. The P-values of All coefficients in the model, including linear terms, quadratic terms, and interaction terms, are less than 0.05, indicating that they are statistically significant at the 95% confidence level.

Table 2. Experimental results for initial preload $V_0$ with two M16 Bolts.

| $N_0$ | Loads $F_i$, N | Experimental Results of $V_{tni}$ under Initial tightening Force $V_0 = 15500$ N | Loads $F_{Vi}$, N | Loads $F_{Hi}$, N | The external load factor ($\chi$) according to Equation (6), with $V_0 = 15500$ N |
|---|---|---|---|---|---|
| 1 | 4000 | 15190 | 0 | 4000 | 0.367 |
| 2 | 4500 | 15278 | 0 | 4500 | 0.371 |
| 3 | 5000 | 15549 | 0 | 5000 | 0.366 |
| 4 | 5500 | 15821 | 0 | 5500 | 0.365 |
| 5 | 6000 | 16120 | 0 | 6000 | 0.373 |
| 6 | 6500 | 16373 | 0 | 6500 | 0.366 |
| 7 | 7000 | 16657 | 0 | 7000 | 0.368 |
| **Average external load factor $\bar{\chi}$** | | | | | **0.368** |

**Table 3. The results the Average external load factor.**

| $N_0$ | Factor symbol | | | Friction coefficient μ | Surface roughness Ra, μm | Material hardness, HB | external load factor $\bar{\chi}$ |
|---|---|---|---|---|---|---|---|
| | $x_1$ | $x_2$ | $x_3$ | μ | Ra | HB | y |
| 1 | −1 | −1 | −1 | 0.1 | 1.6 | 120 | 0.368 |
| 2 | 1 | −1 | −1 | 0.3 | 1.6 | 120 | 0.348 |
| 3 | −1 | 1 | −1 | 0.1 | 6.3 | 120 | 0.337 |
| 4 | 1 | 1 | −1 | 0.3 | 6.3 | 120 | 0.308 |
| 5 | −1 | −1 | 1 | 0.1 | 1.6 | 250 | 0.16 |
| 6 | 1 | −1 | 1 | 0.3 | 1.6 | 250 | 0.132 |
| 7 | −1 | 1 | 1 | 0.1 | 6.3 | 250 | 0.143 |
| 8 | 1 | 1 | 1 | 0.3 | 6.3 | 250 | 0.106 |
| 9 | −1.682 | 0 | 0 | 0.032 | 3.95 | 185 | 0.241 |
| 10 | 1.682 | 0 | 0 | 0.368 | 3.95 | 185 | 0.193 |
| 11 | 0 | −1.682 | 0 | 0.2 | 0.1 | 185 | 0.255 |
| 12 | 0 | 1.682 | 0 | 0.2 | 7.9 | 185 | 0.207 |
| 13 | 0 | 0 | −1.682 | 0.2 | 3.95 | 75.67 | 0.449 |
| 14 | 0 | 0 | 1.682 | 0.2 | 3.95 | 294.33 | 0.104 |
| 15 | 0 | 0 | 0 | 0.2 | 3.95 | 185 | 0.289 |
| 16 | 0 | 0 | 0 | 0.2 | 3.95 | 185 | 0.299 |
| 17 | 0 | 0 | 0 | 0.2 | 3.95 | 185 | 0.295 |
| 18 | 0 | 0 | 0 | 0.2 | 3.95 | 185 | 0.298 |
| 19 | 0 | 0 | 0 | 0.2 | 3.95 | 185 | 0.296 |
| 20 | 0 | 0 | 0 | 0.2 | 3.95 | 185 | 0.295 |

**Table 4. Significance test of regression equation coefficients.**

| Term | Coef | SE Coef | T | P |
|---|---|---|---|---|
| Constant | 0.407 | 0.009659 | 42.131 | 0.000 |
| μ (A) | 1.06 | 0.039788 | 26.743 | 0.000 |
| Ra(B) | 0.024 | 0.001633 | 14.88 | 0.000 |
| HB(C) | $−1.01 \times 10^{-3}$ | 0.000068 | −14.848 | 0.000 |
| μ*μ | −2.78 | 0.065645 | −42.345 | 0.000 |
| Ra*Ra | $−4.14 \times 10^{-3}$ | 0.000119 | −34.805 | 0.000 |
| HB*HB | $−1.88 \times 10^{-6}$ | 0.000345 | −10.299 | 0.000 |
| μ*Ra | $−9.57 \times 10^{-3}$ | 0.003749 | −2.554 | 0.029 |
| μ*HB | $−3.1 \times 10^{-4}$ | 0.000136 | −2.27 | 0.047 |
| Ra*HB | $2.29 \times 10^{-5}$ | 0.000006 | 3.972 | 0.003 |

The friction coefficient (μ) has the strongest influence on the external load factor χ, with a positive regression coefficient (1.06) and a high T-value (26.743). This indicates that as the friction coefficient increases, the reaction force within the threaded joint rises significantly. Surface roughness (Ra) and material hardness (HB) also affect χ, but to a lesser extent. Specifically, HB has a negative effect, meaning that as hardness increases, the external load factor tends to decrease slightly.

The three quadratic terms μ², Ra², and HB² all have negative coefficients and large absolute T-values, indicating clear nonlinear effects. Additionally, the interactions between factors show significant impacts: the interactions μ × Ra and μ × HB

have negative effects, reflecting a decrease in effectiveness when these two factors increase simultaneously. Conversely, the interaction Ra×HB has a positive effect, suggesting that under certain conditions, the combination of surface roughness and hardness may increase the external load factor.

The significance evaluation of these coefficients provides a strong basis for optimizing assembly conditions to effectively control the external load factor in threaded joints.

## Testing the lack of fit in regression

**Analysis of variance.** For testing the lack of fit in regression, the analysis of variance (ANOVA) for significance of regression is shown in Table 5

The adequacy of the regression model was evaluated through the analysis of variance (ANOVA). The results presented in Table 5 show that the model's F-value is 2976.57 with P = 0.0, indicating a very high statistical significance of the model. The Lack-of-Fit test yielded an F-value of 0.01 and P = 1.0, demonstrating no signs of inadequacy in the model, meaning the model fits the experimental data well. The coefficients of determination $R^2$ = 99.96%, adjusted $R^2_{adj}$ = 99.93%, and predicted $R^2_{pred}$ = 99.94% indicate that the model explains almost all the variance in the data and possesses high reliability for prediction. Thus, the obtained regression model is statistically significant, well-fitted with the data, and serves as a reliable tool for analyzing the influence of assembly conditions on the external load factor in bolted joints.

**Evaluation of residuals in the regression model.** Fig 6 presents the diagnostic plots used to assess the residual behavior of the regression model. The normal probability plot shows that the residuals follow an approximately linear trend, indicating that the normality assumption is reasonably satisfied. The small deviation at the lower tail is typical for second-order RSM models and does not represent a systematic pattern.

In the residuals versus fitted values plot, slight deviations are observed at low fitted levels due to the concentration of experiments at low μ; however, no funnel shape or variance trend is present, confirming that the assumption of constant variance holds. The residuals versus observation order plot also shows no noticeable trend or cyclic behavior, demonstrating that the residuals are independent.

Overall, the diagnostic plots confirm that the residuals are randomly scattered without any systematic patterns, and the model satisfies the key assumptions of normality, independence, and homoscedasticity. These results validate the use of the second-order RSM model for prediction, significance analysis, and optimization of the external load factor (χ) within the investigated design space. Consequently, the model is considered reliable for evaluating the influence of technological parameters including surface roughness, friction coefficient and material hardness on load distribution behavior in bolted joints.

## Analysis of the main and interaction effects of input factors

To evaluate the influence of input factors on the output variable in the experimental process, Main Effects Plots were employed, as illustrated in Fig 7. The investigated factors include the Combined friction coefficient (μ), surface roughness

Table 5. Analysis of variance for significance of regression.

| Source | DF | Seq SS | Adj SS | Adj MS | F | P |
|---|---|---|---|---|---|---|
| Regression | 9 | 0.166364 | 0.166364 | 0.018485 | 2976.57 | 0 |
| Residual Error | 10 | 0.000062 | 0.000062 | 0.000006 | | |
| Lack-of-Fit | 5 | 0.000001 | 0.000001 | 0 | 0.01 | 1 |
| Pure Error | 5 | 0.000061 | 0.000061 | 0.000012 | | |
| Total | 19 | 0.166427 | | | | |

R-Sq = 99.96%; R-Sq(pred) = 99.94%; R-Sq(adj) = 99.93%

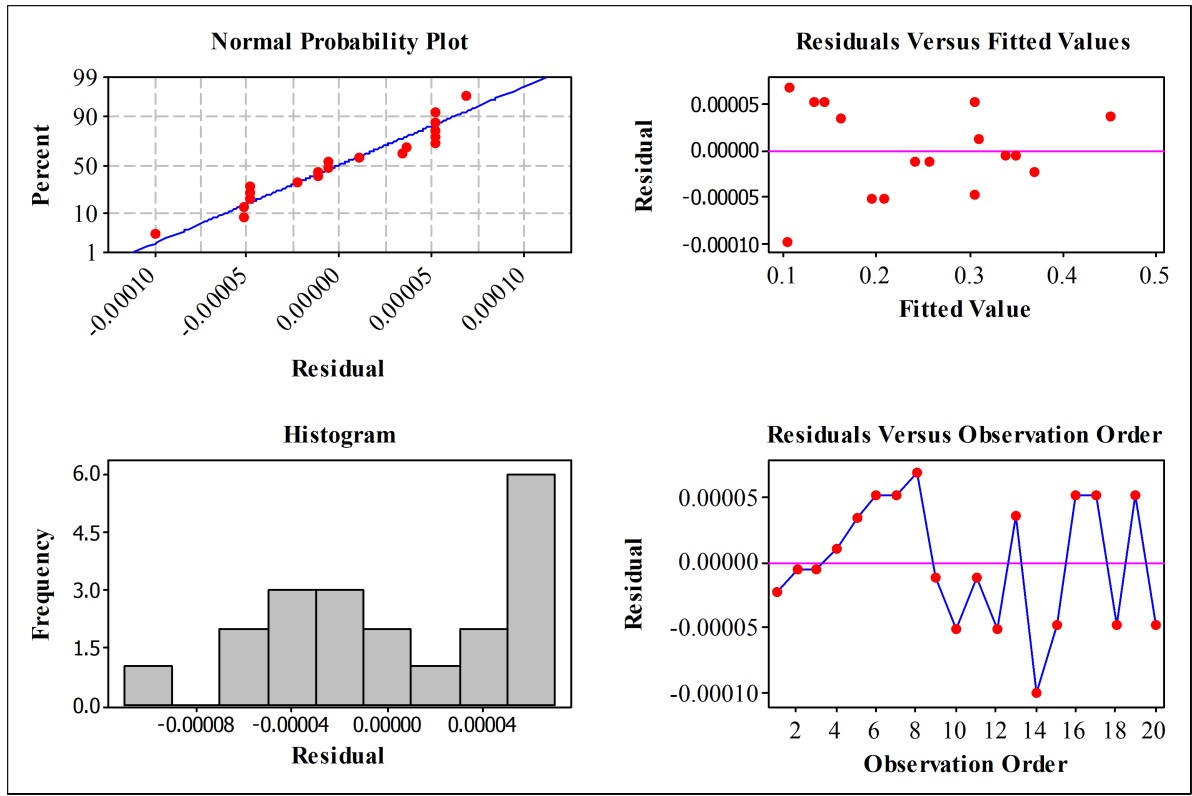

**Fig 6. Diagnostic plots for evaluating the residual behavior of the regression model.**

(Ra), and material hardness (HB) of the joint plates. These plots provide a visual representation of how changes in each individual factor affect the external load factor ($\chi$), while holding other factors constant. This analysis helps to identify which variables have the strongest impact and guides further optimization in assembly design.

**The Combined coefficient of friction**: Fig 7.a illustrates the nonlinear effect of the friction coefficient ($\mu$) on the mean of the external load factor ($\chi$). An increase in $\mu$ from 0.032 to 0.2 leads to a significant rise in the external load factor, reaching a maximum at $\mu = 0.2$. Beyond this point, as $\mu$ increases to 0.368, the external load factor decreases sharply. The steep gradient confirms the sensitivity of the output to friction variations. The red reference line emphasizes the deviation from the overall mean. These results indicate that $\mu \approx 0.2$ is a critical value for optimizing system performance.

**Surface roughness (Ra)**: Fig 7.b illustrates the nonlinear effect of surface roughness (Ra) on the mean of the external load factor ($\chi$). As Ra increases from 0.01 to 3.95 $\mu$m, the external load factor rises, reaching a peak of Ra = 3.95 $\mu$m. Beyond this point, higher roughness levels (6.3 and 7.9 $\mu$m) result in a sharp decline in the external load factor, indicating performance degradation due to excessive surface roughness. The red mean line serves as a reference for deviation of the external load factor. These trends confirm that Surface Roughness (Ra) is a critical parameter, and maintaining roughness near 3.95 $\mu$m is essential for optimal system performance.

**Effect of material hardness**: Fig 7.c presents the effect of material hardness on the mean of external load factor ($\chi$). As hardness increases from 75.68 HB to 294.32 HB, $\chi$ decreases markedly, indicating a strong inverse relationship. The highest $\chi$ is observed at the lowest hardness, while all values at higher hardness levels fall below the global mean (red line). This trend confirms that material hardness negatively affects $\chi$. Although softer materials may enhance $\chi$ under test conditions, practical applications must also consider mechanical strength and wear resistance.

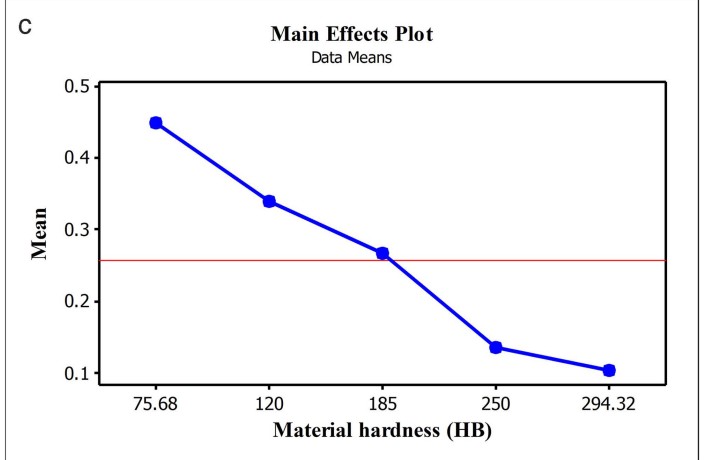

**Fig 7. Main Effects Plot.** (a) Combined friction coefficient; (b) Surface roughness; (c) Material hardness.

In summary, the analysis demonstrates that all three investigated factors significantly affect the external load factor, with clear nonlinear or inverse trends. This underscores the importance of rational design parameter selection to optimize the performance of the system.

## Combined effects of input factors pairs on the external load factor (χ)

Following the development and validation of the regression model for predicting the external load factor (χ), the model was employed to investigate the combined effects of input parameters. Contour and surface plots were used to visualize parameter interactions and identify regions associated with optimal χ values. Unlike single-factor analysis, surface plots reveal optimal operating zones and suitable parameter combinations, enhancing the design reliability of bolted joints. Fig 8 illustrates the joint influence of friction coefficient, surface roughness, and material hardness on χ in bolted connections.

*Effect of μ and Ra on the external load factor χ:* Fig 8.a shows a 3D response surface depicting the combined effect of friction coefficient (μ) and surface roughness (Ra) on the external load factor (χ), based on a second-order RSM model. The surface reveals that χ increases with μ and Ra, reaching a peak before declining, indicating an optimal region. This dome-shaped peak aligns with trends from the main effects plots. The contour map below highlights the optimal zone

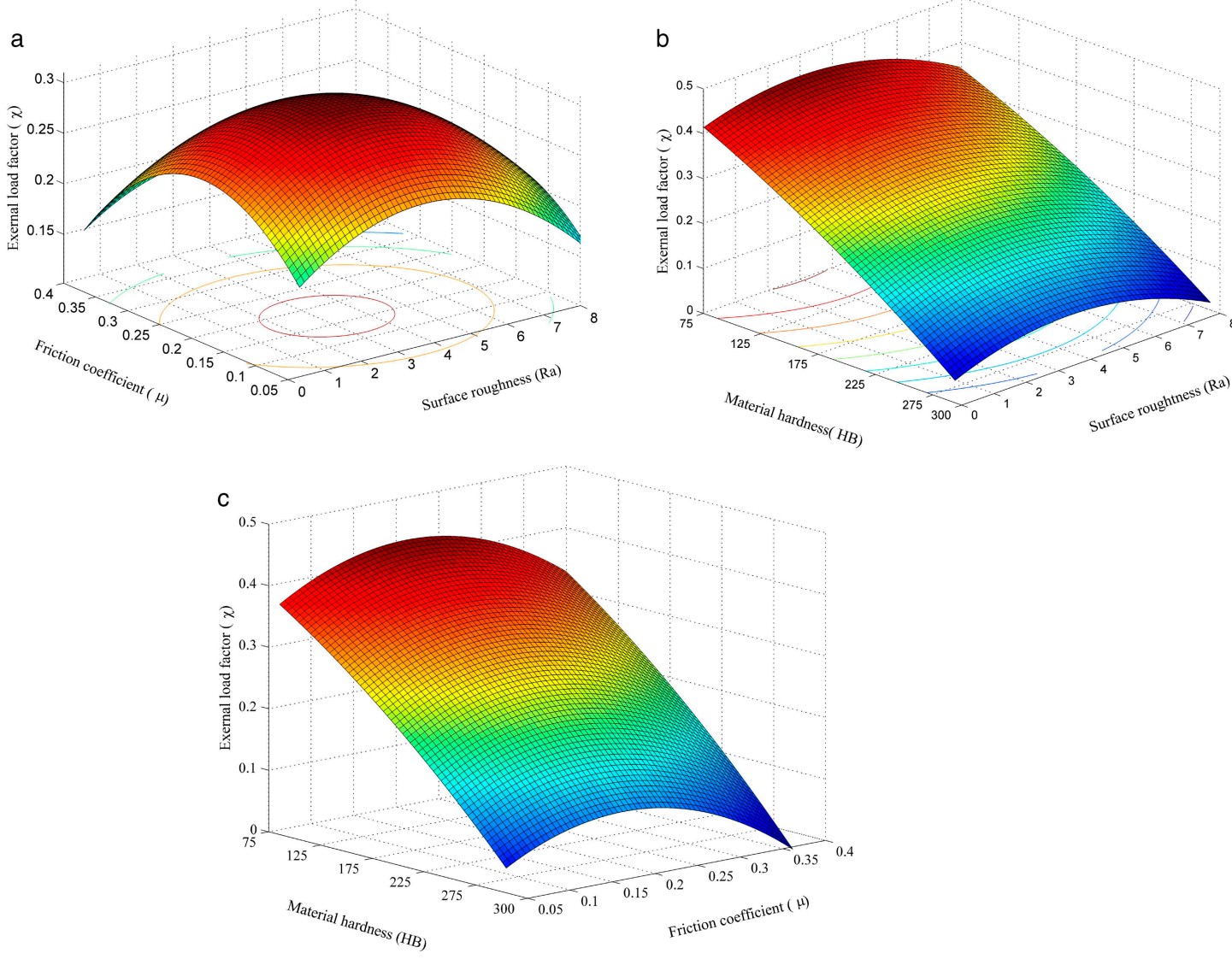

**Fig 8. The effects of input factors on the external load factor.** (a) Effect of µ and Ra on the χ; (b) Effect of HB and Ra on the χ; (c) Effect of HB and µ on the χ.

with red-centered lines. The smooth, symmetric curvature confirms a second-order relationship, validating the model's accuracy. The results emphasize that maximizing χ requires jointly optimizing µ and Ra, clearly demonstrating the value of multi-variable optimization in bolted joint design.

***Effect of HB and Ra on the external load factor χ:*** Fig 8.b presents a 3D response surface showing the combined effect of material hardness (HB) and surface roughness (Ra) on the external load factor (χ). The results indicate that χ decreases sharply as HB increases, especially at low Ra, consistent with previous trends. Softer materials enhance contact deformation, increasing friction and reducing bolt load. At low HB, χ initially rises with Ra, then falls, showing a nonlinear effect. When HB exceeds 250, Ra has minimal influences, as shown by the flattened surface. The contour plot highlights an optimal region at low HB and moderate Ra (3–4 µm), confirming the need for multi-parameter optimization to maximize χ.

***Effect of HB and μ on the external load factor χ:*** Fig 8.c illustrates the combined effect of material hardness (HB) and friction coefficient (μ) on the external load factor (χ). χ decreases linearly with increasing HB (75–300 HB), indicating strong negative influence. μ also reduces χ, with a rapid decline from 0.05 to 0.2, followed by a slower decrease, suggesting a saturation effect. The lowest χ values occur with high HB and high μ, implying minimal load transfer to the bolt. In contrast, soft materials and low friction yield the highest χ. The curved surface indicates nonlinear interaction between HB and μ, confirming that joint optimization requires simultaneous control of both parameters.

## Optimization and sensitivity analysis

**Optimal operating region of the external load factor χ.** The Overlaid Contour Plot tool in Minitab was used to identify the optimal operating region of the input variables, including the coefficient of friction, surface roughness, and hardness of the mating material, where the external load factor (response) falls within the acceptable range of $0.2 < \chi < 0.3$, as illustrated in Fig 9.

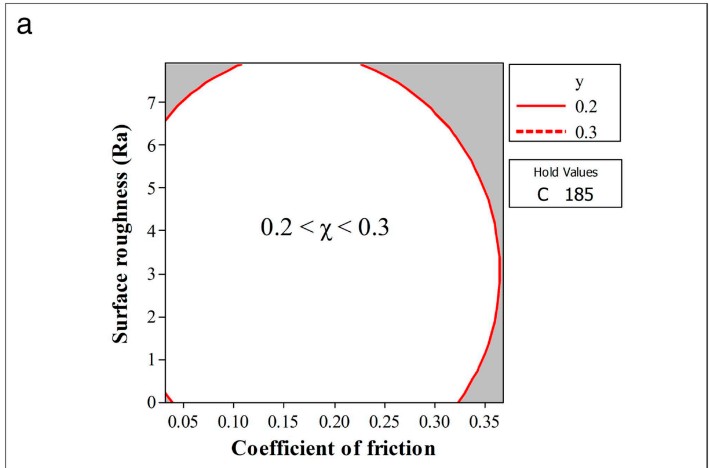

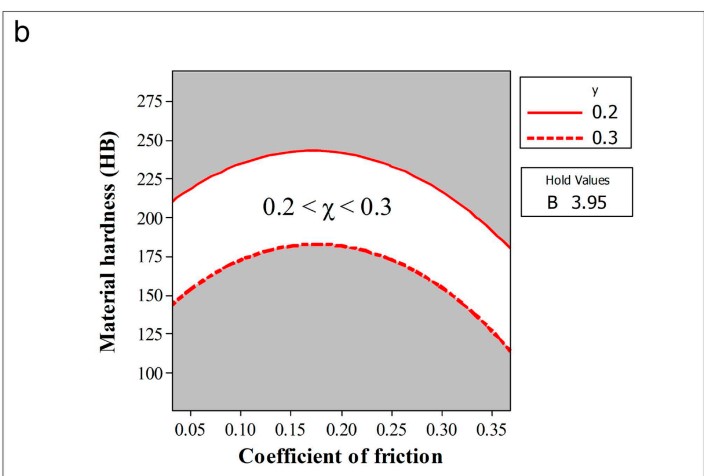

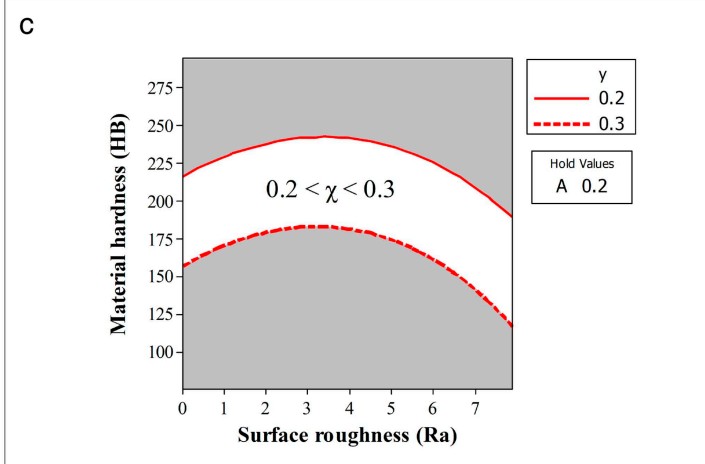

**Fig 9. The operating region of the input factors.** (a) The operating region of μ and Ra; (b) The operating region of μ and HB; (c) The operating region of Ra and HB.

Fig 9a shows a contour plot of the region where the external load factor χ stays within the desired range (0.2 < χ < 0.3), with material hardness fixed at 185 HB. χ is highly sensitive when μ < 0.1 or μ > 0.3, and when Ra is outside 1–6 μm. A stable zone appears for μ ≈ 0.15–0.25 and Ra ≈ 2–5 μm, indicating robust design conditions. The white contour region marks parameter combinations meeting the target χ, while gray areas fall outside. The acceptable zone forms an elliptical shape, reflecting strong interaction between μ and Ra. This insight aids in selecting machining and surface conditions to ensure joint reliability and uniform load distribution.

Fig 9b presents a contour plot showing the effect of friction coefficient (μ) and material hardness (HB) on χ, with Ra fixed at 3.95 μm. The target range (0.2 < χ < 0.3) is met when $0.1 \leq μ \leq 0.3$ and $140 \leq HB \leq 230$. χ is particularly sensitive within the 150–200 HB range, where small changes in μ can push χ beyond acceptable limits. The plot highlights a nonlinear interaction between parameters, with hardness showing a stronger influence than μ. Identifying this safe zone supports optimal material selection and surface condition control, enhancing joint reliability during design and assembly.

Fig 9c shows a contour plot indicating that χ remains within the desired range (0.2 < χ < 0.3) when material hardness is between 170–230 HB and surface roughness between 2–5.5 μm, with μ fixed. The result highlights the dominant role of hardness in controlling load distribution. Materials that are too soft or too hard lead to unstable bolt tension, emphasizing the importance of selecting appropriate hardness and surface finish to ensure joint performance.

The sensitivity analysis shows that maintaining hardness and surface roughness within optimal limits is essential to ensure the effective performance of the joint. The plot indicates high sensitivity of χ to material hardness, especially in the transitional region of 170–230 HB. Even within the stable Ra range (approximately 3–5 μm), small variations in HB can cause χ to exceed the optimal range, emphasizing the need to tightly control HB during machining or material selection.

**The optimal values of the input factors.** The optimization procedure was performed using the desirability-based optimization approach to identify the combination of μ(A), Ra(B), and HB(C) that produces the target external load factor χ(y) ≈ 0.25. The optimal solution obtained from the response surface model corresponds to μ = 0.1809, Ra = 3.5181 μm and HB = 216.859, yielding a predicted value of χ ≈ 0.2501. The composite desirability reaches 0.9996, demonstrating an excellent agreement between the optimization objective and the model prediction, and thus confirming the reliability of the regression model within the design domain.

Fig 10 illustrates the convergence behavior of the optimization process, including the evolution of the predicted χ, the desirability function, and the iterative adjustment of μ, Ra, and HB. This visualization demonstrates that the optimum is achieved through a stable and monotonic convergence path, confirming that the final solution is robust and not highly sensitive to initial conditions. The coordinated adjustment of the three input factors further highlights the nonlinear interaction effects identified in the regression analysis.

Overall, the optimization results establish that the target value χ ≈ 0.25 can be achieved with high confidence, and they validate the practical applicability of the developed regression model for guiding the selection of technological parameters in threaded joint assemblies.

## Conclusions

This study conducted an experimental analysis to evaluate the effects of three technological factors, including combined coefficient of friction, surface roughness, and material hardness of the clamped plate on the external load factor (χ) of threaded joints. A second-order regression model was developed based on a Rotatable Central Composite Design (Box–Wilson) combined with a dedicated measurement system, enabling a quantitative description of the nonlinear behavior and interaction effects among these parameters.

The statistical analysis confirmed that all linear, quadratic, and interaction terms in the model were significant at the 95% confidence level (P < 0.05). The resulting predictive equation exhibited excellent agreement with experimental data ($R^2$ = 99.96%; $R^2_{Adj}$ = 99.93%; $R^2_{Pred}$ = 99.94%), demonstrating the robustness and reliability of the model.

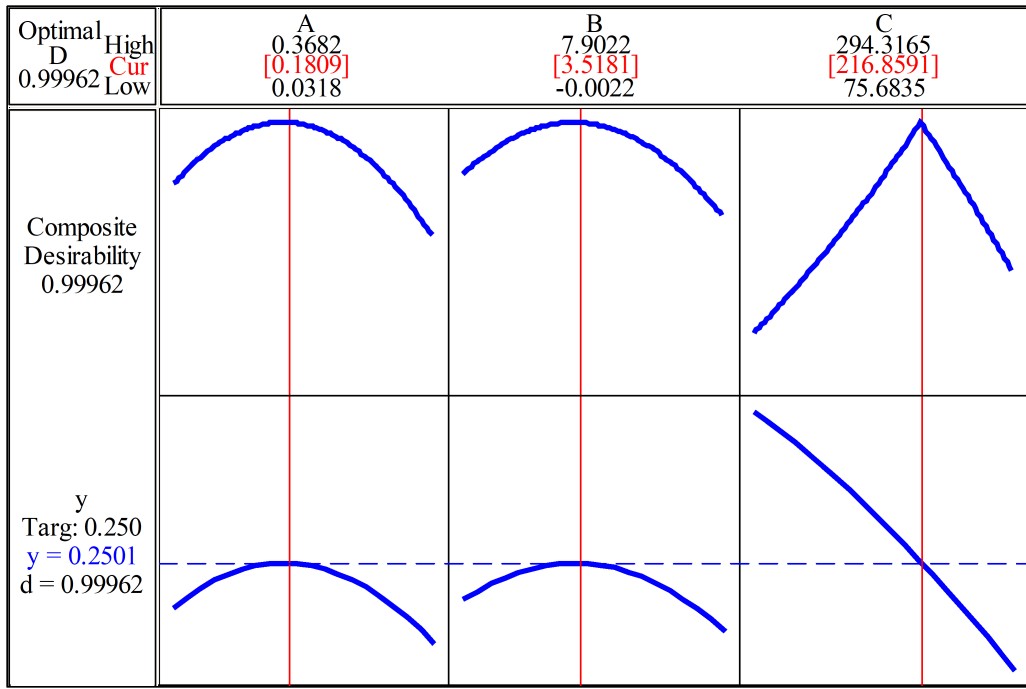

| Optimal D 0.99962 | High Cur Low | A 0.3682 [0.1809] 0.0318 | B 7.9022 [3.5181] -0.0022 | C 294.3165 [216.8591] 75.6835 |
|---|---|---|---|---|
| Composite Desirability 0.99962 | | | | |
| y Targ: 0.250 y = 0.2501 d = 0.99962 | | | | |

**Fig 10. Convergence behavior of the response surface optimization process.**

The regression results indicated that friction coefficient (μ) had the strongest influence (linear coefficient +1.06; quadratic coefficient –2.78), producing a nonlinear trend with a maximum at μ ≈ 0.2. Surface roughness (Ra) also exhibited a pronounced nonlinear effect (linear coefficient +0.024; quadratic coefficient –4.14 × 10$^{-3}$), with a maximum at Ra ≈ 3.9 μm. Material hardness (HB) caused a stable decrease in the external load factor (χ) over the entire range studied (linear coefficient –1.01 × 10$^{-3}$).

The evaluation of the relative contributions based on the total absolute T-value showed that μ accounted for ≈46%, Ra ≈ 35%, and HB ≈ 19% of the variation in the external load factor (χ). The Ra × HB interaction was the strongest among the pairs, indicating a significant dependence of roughness effects on material hardness. These results are consistent with the analyses presented in the discussion and reinforce the comprehensiveness of the regression model.

Response Optimizer predicted χ ≈ 0.2501 at the optimal combination μ = 0.1809, Ra = 3.5181 μm, HB = 216.859, with a total desirability of 0.9996, indicating high predictive and optimization capability. Overall, this study provides a reliable quantitative framework for controlling the external load factor, directly contributing to the enhanced reliability of threaded joints in engineering applications.

### Future research directions

The quantitative results obtained from the Box–Wilson model open several potential directions for future research. First, the model may be extended by incorporating additional geometric and mechanical variables suggested by Buckingham's π-theorem to provide a more comprehensive description of the behavior of threaded joints. Second, the statistically significant regression coefficients offer a solid basis for implementing reliability-based design optimization (RBDO) to control the external load factor χ under uncertain operating conditions. Furthermore, a standardized laboratory benchmark for load testing should be developed to evaluate various joint configurations and validate the model's applicability on a broader scale. These directions arise directly from the core scientific contributions of the present study.

## Supporting information

**S1 File. Experimental data for the regression equation (22).**
(RAR)

## Acknowledgments

We acknowledge the support of time and facilities from Pham Van Dong University and Ho Chi Minh City University of Technology (HCMUT), VNU-HCM for this study.

## Author contributions

**Conceptualization:** Van Thuy Tran.

**Data curation:** Van Thuy Tran.

**Formal analysis:** Van Thuy Tran.

**Funding acquisition:** Van Thuy Tran.

**Investigation:** Van Thuy Tran, Huu Loc Nguyen.

**Methodology:** Van Thuy Tran.

**Project administration:** Van Thuy Tran.

**Resources:** Van Thuy Tran.

**Software:** Van Thuy Tran.

**Supervision:** Van Thuy Tran, Huu Loc Nguyen.

**Validation:** Van Thuy Tran, Huu Loc Nguyen.

**Visualization:** Van Thuy Tran, Huu Loc Nguyen.

**Writing – original draft:** Van Thuy Tran, Huu Loc Nguyen.

**Writing – review & editing:** Van Thuy Tran, Huu Loc Nguyen.

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
