## [Decision Letter · Decision Letter 0]

13 Oct 2025

Dear Dr. Tran,

We look forward to receiving your revised manuscript.

Kind regards,

Giulia Pascoletti, Ph.D.

Academic Editor

PLOS ONE

Journal Requirements:

2. We note that your Data Availability Statement is currently as follows: “All relevant data are within the manuscript and its Supporting Information files.”

3. When completing the data availability statement of the submission form, you indicated that you will make your data available on acceptance. We strongly recommend all authors decide on a data sharing plan before acceptance, as the process can be lengthy and hold up publication timelines. Please note that, though access restrictions are acceptable now, your entire data will need to be made freely accessible if your manuscript is accepted for publication. This policy applies to all data except where public deposition would breach compliance with the protocol approved by your research ethics board. If you are unable to adhere to our open data policy, please kindly revise your statement to explain your reasoning and we will seek the editor's input on an exemption. Please be assured that, once you have provided your new statement, the assessment of your exemption will not hold up the peer review process

Additional Editor Comments:

**Please, note that, due to some technical issues, Reviewer 1 has modified the initial review report, and you can find it attached to this email. Refer to this version of the report only for Reviewer 1.**

**Comments to the Author**

1. Is the manuscript technically sound, and do the data support the conclusions?

Reviewer #1: Yes

Reviewer #2: Partly

2. Has the statistical analysis been performed appropriately and rigorously?

Reviewer #1: Yes

Reviewer #2: No

3. Have the authors made all data underlying the findings in their manuscript fully available?

Reviewer #1: Yes

Reviewer #2: Yes

4. Is the manuscript presented in an intelligible fashion and written in standard English?

Reviewer #1: Yes

Reviewer #2: Yes

Reviewer #1: The manuscript presents a clear and well-structured experimental study aimed at evaluating how friction coefficient, surface roughness, and material hardness influence the external load factor (χ) in bolted joints. The topic is relevant for structural and mechanical engineering, and the experimental approach is sound. The text is well-organized, and the theoretical background is adequately introduced.

However, from a technical standpoint, the absence of figures and tables (which are referenced throughout the text) makes it difficult to fully assess the quality, reliability, and relevance of the experimental results and their interpretation. The visual data presentation (e.g., test setups, regression plots, material properties, result distributions) is crucial for a complete evaluation of the work.

External Load Factor Definition and Use

The definition and calculation of the external load factor (χ) are clearly introduced. However, consistency in terminology is recommended — the manuscript sometimes uses "external force coefficient" interchangeably with "external load factor". To avoid confusion, use a single term throughout, ideally aligned with standard nomenclature.

Experimental Parameters and Materials

The choice of parameters — friction coefficient, roughness, and hardness — is appropriate and well-justified. The method used to modify surface conditions is mentioned, but without the accompanying images, it's difficult to judge the degree of control over surface quality and treatment reproducibility.

Test Setup and Procedure

The test configuration and measurement procedures are mentioned but not fully detailed without the figures. Key information such as bolt dimensions, tightening method, load application system, and instrumentation setup (e.g., load cells, strain gauges, torque sensors) are critical for evaluating the robustness of the experimental design.

Discussion of Results

The qualitative discussion is promising, suggesting that roughness and hardness are significant contributors to joint performance. Yet, without access to the quantitative figures or comparative charts (as cited in the manuscript), the strength of the conclusions remains difficult to validate.

Reviewer #2: This study experimentally investigates how friction coefficient, surface roughness, and material hardness of clamped plates influence the external load factor (χ) in threaded joints. Results show that material hardness has the strongest effect, with χ decreasing as hardness increases, while surface roughness has a nonlinear impact and friction coefficient also plays a significant role. A regression model developed using the Box–Wilson method accurately predicts χ, with an optimization identifying the best parameter combination for reliable joint performance. The findings provide practical insights for improving the design and reliability of bolted joints in engineering applications. While the paper appears to contribute meaningfully to the state of the art, the overall integrity of the manuscript is questionable. The text is difficult to follow, and the terminology used is challenging for readers, even for experts in the field. I believe a comprehensive revision of both the text and structure (potentially a polished, rewritten version) is necessary before the manuscript can be considered for possible publication. The main points for improvement are as follows:

Comment No. 1: The Abstract and Introduction are reasonably well-prepared. However, the content in Section 2 is unclear, and many terms are vaguely defined. For example, it is not evident what is meant by the “YY plane” (is this the 2D plane where equilibrium has been illustrated in Fig. 1?), or what is referred to as the “reference axis” (is this the trimmed line?). Similarly, the centroid point O, introduced in another section cut, needs clarification regarding its relation to the upper section (is it a vertical section from the Y-direction main vector plane?). All figures should be revised to provide sufficient detail for comprehension rather than requiring the reader to decode them.

Comment No. 2: It is strongly advised that the authors present all external and internal forces acting on the system to improve clarity. The terms defined in Eq. (1) are not visible in Fig. 1, making their definition and relevance to the study ambiguous. Definitions of variables such as Fb and Fm1 are also missing. Section 2 must be rewritten with adequate explanations of the equations and free-body equilibrium diagrams to ensure reader understanding.

Comment No. 3: In Section 3, the study seems to describe a system designed for a special purpose based on mathematical assumptions. The order of Figs. 1, 2, and 3 should be revised, with corresponding explanations currently in Section 3 moved to Section 2.

Comment No. 4: Section 3.2 suggests that surface roughness directly relates to friction. If this is the case, the two factors cannot be considered independent variables. The authors need to justify why both were included in their analysis and equations.

Comment No. 5: The quality of figures and tables must be improved throughout the manuscript. Figure 5 appears redundant as it repeats content already shown elsewhere, with little justification for its re-presentation. Descriptions within figures and in the main text are insufficient, making interpretation difficult and undermining the paper’s integrity and quality. Additionally, if no “Note” exists in Table 4, it should be removed.

Comment No. 6: A clear explanation is required on how the dataset for Fig. 7 was obtained, how the data were summarized in Table 5, and whether the results were derived experimentally, numerically, or through another approach. The rationale for applying linear equations without considering dimensional analysis or the Buckingham π-theorem is unclear. The manuscript must also explain whether the Box–Wilson method adequately satisfies these theoretical requirements.

Comment No. 7: The final paragraph of the Conclusion does not present conclusions but rather a potential application of the study; it should either be moved to another section or removed entirely. Moreover, the regression results and the effects of each factor, discussed earlier in the paper, are missing from the Conclusion. The most significant achievement appears to be the relationships among the factors, their percentage contributions, and the coefficients obtained from the Box–Wilson method, and this should be emphasized.

Comment No. 8: The final paragraph of the Conclusion does not present conclusions but rather a potential application of the study; it should either be moved to another section or removed entirely. Moreover, the regression results and the effects of each factor, discussed earlier in the paper, are missing from the Conclusion. The most significant achievement appears to be the relationships among the factors, their percentage contributions, and the coefficients obtained from the Box–Wilson method, and this should be emphasized.

Comment No. 9: Only 9 out of 19 cited references are from the past five years. The literature review should be updated with more recent and relevant works to strengthen the paper’s contribution to the state of the art.

Comment No. 10: Finally, while the proposed future directions add some value, they would be more compelling if linked explicitly to the paper’s contribution to the field. For example: Did the study omit certain influential factors identified through Buckingham’s π-theorem and dimensional analysis? Was reliability analysis, as outlined in Section 4, neglected? Was a laboratory-scale benchmark for load testing not designed? Regardless of the answers, the paper must clearly explain how this study differentiates itself from previous research and what unique contribution it offers.

**Do you want your identity to be public for this peer review?** For information about this choice, including consent withdrawal, please see our Privacy Policy

Reviewer #1: No

Reviewer #2: **Yes: ** Elshan Ahani

---

## [Author Response · Author response to Decision Letter 1]

18 Nov 2025

Please see the attached Response to Reviewers document for detailed replies to all comments

---

## [Decision Letter · Decision Letter 1]

16 Dec 2025

Experimental Investigation of the Influence of Friction, Surface Roughness and Material Hardness on the External Load Factor in Threaded Joints

PONE-D-25-35836R1

Dear Dr. Tran,

We’re pleased to inform you that your manuscript has been judged scientifically suitable for publication and will be formally accepted for publication once it meets all outstanding technical requirements.

Kind regards,

Giulia Pascoletti, Ph.D.

Academic Editor

PLOS One

Additional Editor Comments (optional):

Reviewers' comments:

Reviewer's Responses to Questions

**Comments to the Author**

Reviewer #1: All comments have been addressed

Reviewer #2: All comments have been addressed

2. Is the manuscript technically sound, and do the data support the conclusions?

Reviewer #1: Yes

Reviewer #2: Yes

3. Has the statistical analysis been performed appropriately and rigorously?

Reviewer #1: Yes

Reviewer #2: Yes

4. Have the authors made all data underlying the findings in their manuscript fully available?

Reviewer #1: Yes

Reviewer #2: Yes

5. Is the manuscript presented in an intelligible fashion and written in standard English?

Reviewer #1: Yes

Reviewer #2: Yes

Reviewer #1: (No Response)

Reviewer #2: I think the improvements are sufficient to recommend the paper for publication. The manuscript is now clear, well structured, and technically sound.

**Do you want your identity to be public for this peer review?** For information about this choice, including consent withdrawal, please see our Privacy Policy

Reviewer #1: No

Reviewer #2: **Yes: ** Elshan Ahani

---

## [Editor Report · Acceptance letter]

PONE-D-25-35836R1

PLOS One

Dear Dr. Tran,

I'm pleased to inform you that your manuscript has been deemed suitable for publication in PLOS One. Congratulations! Your manuscript is now being handed over to our production team.

Kind regards,

on behalf of

Dr. Giulia Pascoletti

Academic Editor

PLOS One